# Evaluating Apelin as a Potential Biomarker in Major Depressive Disorder: Its Correlation with Clinical Symptomatology

**DOI:** 10.3390/ijms252413663

**Published:** 2024-12-20

**Authors:** Enkhmurun Chibaatar, Rintarou Fujii, Atsuko Ikenouchi, Naomichi Okamoto, Tomoya Natsuyama, Gaku Hayasaki, Takahiro Shinkai, Reiji Yoshimura

**Affiliations:** 1Department of Psychiatry, University of Occupational and Environmental Health, Kitakyushu 807-8555, Japan; enkhmurun@med.uoeh-u.ac.jp (E.C.);; 2Wakamatsu Hospital, University of Occupational and Environmental Health, Kitakyushu 808-0024, Japan

**Keywords:** serum apelin, plasma apelin, major depressive disorder, MADRS, peripheral biomarker

## Abstract

To date, only a limited number of studies have investigated the potential effects of apelin on mood regulation and emotional behavior. Therefore, this study investigated apelin’s role in major depressive disorder (MDD) by comparing the serum and plasma apelin concentrations between 30 patients with MDD and 30 healthy controls (HCs), and the correlated serum and plasma apelin levels and the severity of depressive symptoms using the Montgomery–Åsberg Depression Rating Scale (MADRS). Blood samples were collected following 12 h of fasting, and the apelin levels were measured using an ELISA kit. The serum apelin concentrations showed no significant difference between the MDD and HC groups, while the plasma apelin levels were significantly lower in the MDD group (*p* = 0.002). Among the patients with MDD, a positive moderate correlation was observed between the total MADRS scores and plasma apelin levels (r = 0.439), with statistical significance (*p* < 0.05). Additionally, significant positive correlations (*p* < 0.05) were found between both the serum and plasma apelin levels and the MADRS subscales 5 (reduced appetite) and 6 (concentration difficulties). These preliminary findings, although not definitive, suggest that apelin profiles may help to identify distinct subgroups within MDD patients, warranting further investigation into the different apelin isoforms and their associations in different populations of MDD patients.

## 1. Introduction

Major depressive disorder (MDD) is a prevalent affective disorder characterized by significant alterations in mood, cognition, and behavior. It affects approximately 300 million people worldwide and is the largest contributor to global disability [1,2]. Pathologically, it is characterized by a range of structural and functional alterations in the brain, including neurotransmitter imbalances [3,4], neuroendocrine dysregulation [5,6], neuroinflammation [7,8], and oxidative stress [9,10]. Furthermore, genetic predispositions [11,12] and environmental stressors [13] significantly affect these pathological changes. In addition, hypothalamus–pituitary–adrenal (HPA) axis dysregulation is a critical component of the pathophysiology of MDD, as it mediates the stress response and may perpetuate depressive symptoms through sustained neuroendocrine disruption [14]. In recent years, there has been growing interest in identifying the peripheral biomarkers that could enhance our understanding of the pathophysiology of MDD and facilitate accurate diagnosis, rapid treatment initiation, and effective disease monitoring [2].

Apelin, an endogenous peptide first isolated in 1998 from the bovine stomach by Fujino et al. [15], has emerged as an important neuropeptide in animals and humans. Its biological action is mediated by the high-affinity single G protein-coupled apelin receptor (APJ). Apelin has multiple endogenous bioactive isoforms, ranging from 13 to 55 amino acids, exerting a variety of important functions in both physiological and pathological conditions [16]. Apelin and APJ are extensively expressed in several peripheral organs, including the heart, kidneys, lungs, and stomach [17,18,19,20,21,22], where they positively influence important physiological processes, such as cardiovascular regulation, fluid homeostasis, energy metabolism, and neuroendocrine function [23]. In particular, apelin has attracted interest as a potential biomarker of neuroprotection due to its diverse physiological roles and wide expression in the central nervous system (CNS) [24]. It is expressed in various parts of the CNS, including the hippocampus, thalamus, hypothalamus, amygdala, pituitary gland, and spinal cord [25]. Notably, animal and human studies have reported varying functions of apelin in the CNS. Previous studies have shown that apelin significantly mitigates neurological deficits, reduces infarct volume, and protects the blood–brain barrier (BBB) in rodent models of cerebral ischemia [26,27]. Furthermore, this peptide has been demonstrated to decrease neuronal damage in mice subjected to a traumatic brain injury [28]. Reportedly, it is a promising target for neurodegenerative diseases, including Parkinson’s and Alzheimer’s disease, in both animals and humans [29,30,31].

The potential effects of apelin on mood regulation and emotional behavior have been investigated, but the studies conducted so far are limited. Specifically, the intracerebroventricular (ICV) administration of apelin-13 in rats was shown to reverse depression-like behaviors induced by chronic stress models, including social defeat and unpredictable chronic mild stress (UCMS) [32,33]. Furthermore, recognition memory was improved, and antidepressant-like effects were produced [34]. Numerous studies have suggested that apelin may exert antidepressant-like effects, potentially through the modulation of the hypothalamus–pituitary–adrenal (HPA) axis [35,36]. However, the precise mechanisms underlying these effects remain poorly understood, and the findings regarding apelin concentrations in the blood remain inconsistent across clinical studies.

Within this framework, we aimed to achieve two primary objectives in our study. First, we sought to investigate the differential levels of apelin in the serum and plasma of healthy controls (HCs) and patients with MDD. This approach was informed by the variability in prior research, where apelin levels have predominantly been studied in either the serum or plasma, with limited investigations addressing both matrices simultaneously. Additionally, the studies that have analyzed both the serum and plasma have typically focused on specific apelin isoforms, such as apelin-13 and apelin-36, rather than the total apelin levels. Given the limited understanding of the roles of specific isoforms, assessing the total apelin in both the serum and plasma could provide a valuable measure of the systemic apelin concentration in patients with MDD. This dual analysis allowed for the cross-validation of the findings and facilitated a more comprehensive evaluation by capturing the combined effect of all bioactive isoforms on apelin receptors. By examining both matrices, this study aimed to provide a robust understanding of apelin’s biological distribution and its potential as a biomarker in mood disorders. Second, we sought to elucidate the association between the total serum and plasma apelin levels and the severity of depressive symptoms using the Montgomery–Åsberg Depression Rating Scale (MADRS), a standardized measure of depressive symptomatology. This aspect of our study might reveal preliminary associations in MDD and allow for the exploration of the potential predictive value of total apelin levels with respect to the presence of depressive symptoms as well as to their degree of severity, thereby providing insights that could guide researchers towards considering apelin as a biomarker in clinical settings.

## 2. Results

This study’s sample comprised 60 participants. Approximately 30 (14 male, 16 female) were diagnosed with MDD and another 30 (14 male, 16 female) were the HCs. The mean age was 44.50 ± 11.717 years for the patients with MDD, and 41.47 ± 9.801 years for the HCs. The mean BMI was 24.24 ± 3.52 kg/m^2^ in the patient group and 23 ± 2.44 kg/m^2^ in the HCs. The two groups, regarding age (*p* = 0.281), sex ratio (*p* = 0.258), and body mass index (BMI) (*p* = 0.119), did not differ significantly. Detailed comparisons of the demographic data are summarized in Table 1.

**Serum and plasma concentration of apelin:** The serum and plasma apelin levels in the patients with MDD and the HCs were measured using a solid-phase competitive ELISA. The mean serum apelin concentration of the patients with MDD was 1.694 ± 0.684 ng/mL, and 1.882 ± 0.673 ng/mL in the HCs. The two groups did not differ significantly (*p* = 0.293). In contrast, the mean plasma apelin concentration was significantly lower in the patients with MDD (2.185 ± 0.547 ng/mL) than in the HCs (2.587 ± 0.411 ng/mL), with a *p*-value of 0.002, indicating statistical significance (Table 2). A graphical representation of the same data is provided in Figure 1, illustrated as a scatterplot of the individual values. Moreover, as reported in Table 1, the clinical profile of the patients in this study indicates a moderate severity of depressive symptoms.

**Correlations between apelin concentration and MADRS scores:** We also examined the correlation between apelin levels (both in the serum and plasma) and the MADRS scores in the patients with MDD. The detailed partial correlation data are provided in Table 3 and Figure 2. A moderate positive correlation was observed between the total MADRS score and apelin levels in the plasma (r = 0.439), with statistical significance (*p* < 0.05). In addition, significant positive correlations were observed for the MADRS 5 (reduced appetite) and MADRS 6 (concentration difficulties), among the individual MADRS items (Figure 2).

Specifically, the MADRS 5 (reduced appetite) scores were positively correlated with the serum (r = 0.499, *p* < 0.05) and plasma (r = 0.52, *p* < 0.05) apelin levels. Similarly, the MADRS 6 (concentration difficulties) scores showed significant positive correlations with the serum (r = 0.494, *p* < 0.05) and plasma (r = 0.567, *p* < 0.05) apelin levels (Figure 3). The correlations between the apelin levels and the other MADRS items were generally positive; however, they were statistically nonsignificant, with r-values ranging from 0.009 to 0.378.

## 3. Discussion

To the best of our knowledge, this is the first preliminary clinical study to quantify the total apelin concentrations in both serum and plasma samples from patients with MDD and to compare these levels with those observed in HCs. Our findings indicate that the apelin concentrations in both the plasma and serum were significantly lower in the patients with MDD compared to the HCs. Notably, while the plasma apelin levels showed a statistically significant difference between the two groups, the serum apelin concentrations did not exhibit a significant discrepancy. Additionally, a moderate positive correlation was found between the plasma apelin levels and the total scores on the MADRS. Furthermore, both the plasma and serum apelin concentrations were positively correlated with specific subscales of the MADRS, namely items 5 (reduced appetite) and 6 (concentration difficulties), in the patients with MDD.

Our study contributes to the growing body of literature on the role of apelin in depression. While some studies have found elevated apelin levels in patients with MDD, others have reported reduced levels. The discrepancies between studies underscore the complexity of apelin’s role in MDD and highlight the need for further exploration. Consistent with some prior studies, Puşuroğlu et al. [37] reported significantly lower plasma apelin levels in individuals diagnosed with MDD. Additionally, a study in adolescents with MDD found significantly lower serum apelin levels compared to healthy controls [38]. These findings suggest that apelin may be reduced in certain populations with depression, potentially indicating a dysregulation of apelin in the pathophysiology of the disorder. On the other hand, several studies have reported increases in apelin concentrations in depressed patients. Oguz et al. found that patients with higher Beck Depression Inventory scores exhibited elevated serum apelin-12 levels [39]. Similarly, in another clinical study, serum apelin levels were found to be significantly higher in patients diagnosed with depression compared to healthy controls. No significant changes in apelin levels were observed after 3 months of treatment, despite the clinical recovery from depressive symptoms [40]. In contrast to these studies, our results show a significant reduction in plasma apelin levels in the patients with MDD, suggesting a potential link between blunted apelin release and MDD. This finding is intriguing, since the serum apelin levels, despite being lower on average in the MDD patients, did not exhibit significant differences between the two groups under investigation. This suggests that plasma measurements might be more reflective of the biological changes underlying the pathophysiology of MDD. The discrepancies between the serum and plasma apelin levels may have arisen from differences in sample collection. Serum is obtained after clotting, which can trigger the release of bioactive molecules or influence the removal of apelin by platelets or clotting factors. In contrast, plasma measurements, taken before clotting, may provide a more accurate reflection of the circulating apelin, potentially offering better insight into its role in MDD pathophysiology. However, further research is needed to confirm the precise mechanisms behind these differences. Moreover, the observed differences between our findings and those of previous studies may reflect the variability in apelin levels reported across different studies, which may be influenced by factors such as study design, population demographics, and the specific apelin variant measured. For example, in peritoneal dialysis patients with depression and anxiety, serum apelin levels were significantly higher than in those without depression and anxiety [39]. Moreover, Bullich et al. [41] observed elevated plasma apelin levels, particularly in older individuals with depression. These contrasting findings highlight the complexity of apelin’s physiological role and suggest that variations in this peptide in the bloodstream may reflect different depressive conditions. Further investigation with more detailed clinical presentations of the patients is needed to better understand the relationship between circulating apelin fluctuations and MDD symptomatology.

The correlation analysis carried out in the patients with MDD revealed a significant positive association between reduced plasma apelin levels and more severe depressive symptoms, as measured by the total MADRS score. This result is intriguing, as it contrasts with previous research by Bullich et al. [41], which reported a positive correlation between higher plasma apelin levels and more severe depressive symptoms in older adults with depressive symptoms. These findings also align with an earlier animal study, which demonstrated that administering apelin-13 to the CNS induced depression-like behavior in mice through the APJ receptor [42]. However, our study’s findings suggest that lower plasma apelin levels are associated with greater depression severity, indicating a potential inverse relationship between apelin levels and depression in our sample. This positive correlation between reduced apelin levels and more severe depressive symptoms may initially seem counterintuitive, as many studies have suggested that lower apelin levels are associated with depression. However, it is possible that this finding reflects a more complex biological response. Specifically, this positive correlation could indicate that in cases of more severe depression, apelin might be involved in a compensatory or adaptive mechanism aimed at mitigating certain symptoms. 

Additionally, the serum apelin levels were not significantly correlated with the MADRS scores in our study, further suggesting that plasma apelin may be a more accurate marker of depressive symptoms compared to serum apelin. This contrasts with a previous study that reported a correlation between serum apelin levels and depression and anxiety scores [39]. These contrasting findings highlight the complexity of the role of apelin in depression and underscore the need for further research to clarify how plasma and serum apelin levels influence depressive states differently, and to determine which is a more accurate biomarker for mood disorders.

Our study further revealed a significant positive association between plasma and serum apelin levels and specific depressive symptoms, notably MADRS 5 and MADRS 6, which are associated with reduced appetite and concentration difficulties, respectively [43]. These findings suggest that apelin could play a role in regulating appetite and attentional difficulties in patients with MDD. Specifically, lower plasma apelin levels were associated with more severe reductions in appetite, suggesting that apelin influences the appetite through its neurobiological effects. This is consistent with apelin being involved in modulating the energy balance and feeding behavior. Previous research has shown that apelin interacts with the hypothalamic pathways that regulate appetite and energy homeostasis. This is further supported by studies indicating that apelin-13 administration induces a depression-like phenotype in mice [44], and that lower levels of apelin are found in individuals with anorexia nervosa [45]. Furthermore, apelin may affect the cognitive functions associated with MDD, including attention and concentration difficulties. Therefore, the underlying pathophysiology may be related to the influence of apelin on neurocognitive function through its interaction with neurotransmitter systems and neuroinflammatory processes. Previous studies have explored the role of apelin in cognitive regulation, with evidence linking plasma apelin-13 levels to cognitive impairments, including attention-deficit/hyperactivity disorder (ADHD) and environmental factors influencing cognitive function [46,47]. Furthermore, these findings underscore the broader role of apelin in the pathophysiology of MDD, affecting both physical symptoms such as appetite and cognitive domains, due to its interactions with neurotransmitter systems and neurobiology. Therefore, further investigations are needed to completely explain the role of apelin in the pathophysiology of MDD and its potential as a critical mediator in the interplay between mood disorders and related symptoms.

Despite this study’s contributions, some limitations must be acknowledged. First, the relatively small sample size may limit the generalizability of our findings. Second, the cross-sectional design of this study restricted our ability to establish causality between apelin levels and depressive symptoms. Future studies should focus on larger longitudinal designs to validate and extend these results. Additionally, this study relied on peripheral samples, which may not fully capture apelin levels in the CNS. Finally, measuring the total apelin levels alone without differentiating between subtypes may obscure important variations. Future studies should consider these factors and explore the role of apelin in different depression subtypes and its interactions with other biomarkers to provide a more comprehensive understanding of its clinical potential.

## 4. Materials and Methods

Ethical statement: This study was conducted following the Declaration of Helsinki and adhered to the Japanese Ethical Guidelines for Medical and Health Research Involving Human Subjects. This study’s protocol involving human participants was approved by the Ethics Committee of the University of Occupational and Environmental Health (UOEH), Japan (approval code: UOEHCRB21-164; approval date: 19 January 2022). Study population: Two groups of participants were enrolled in this study: 30 patients with MDD and 30 HCs. Patients with MDD were recruited from both inpatient and outpatient settings at the UOEH hospital. The diagnosis was confirmed by trained psychiatrists using the standard criteria from the *Diagnostic and Statistical Manual of Mental Disorders-5* (DSM-5) [48]. Symptoms were assessed using MADRS, a clinician-administered instrument widely used to assess the severity of depressive symptoms in patients with MDD. It comprises 10 items, each evaluating a distinct symptom domain associated with depression. Each item is scored on a 7-point scale, ranging from 0 (absence of symptoms) to 6 (severe symptoms), with a total score ranging from 0 to 60. Higher total scores reflect greater severity of depressive symptoms [49,50]. The 10 items are described as follows: (1) Apparent sadness: observable despondency, gloom, and despair, assessed through speech, facial expression, and posture. (2) Reported sadness: subjective reports of low mood and hopelessness, rated by intensity and duration. (3) Inner tension: feelings of discomfort, edginess, or mental tension, escalating to panic or dread. (4) Reduced sleep: decreased sleep duration or quality compared to the individual’s usual pattern. (5) Reduced appetite: loss of desire for food, requiring effort to maintain regular eating. (6) Concentration difficulties: impairment in focusing or maintaining attention. (7) Lassitude: difficulty initiating and performing routine activities due to low energy. (8) Inability to feel: reduced emotional responsiveness and diminished interest in normally pleasurable activities. (9) Pessimistic thoughts: negative cognitions, including guilt, self-reproach, and feelings of worthlessness. (10) Suicidal thoughts: thoughts of death, suicidal ideation, or preparation for suicide. Patients included in this study met the following criteria at baseline: (1) aged 20 to 64 years at the time of enrollment, regardless of gender; (2) diagnosed with MDD according to DSM-5 criteria, with a duration of symptoms between 4 and 12 weeks; (3) no history or comorbidity of neurodegenerative disorders (e.g., dementia) or other psychiatric conditions aside from MDD; and (4) absence of serious comorbidities requiring inpatient treatment. There were no restrictions regarding psychotropic medication use or the frequency of hospital visits for outpatient participants. However, to account for variations in psychotropic treatment and ensure standardization of data analysis, we utilized the Japanese guidelines for antidepressant drug equivalence. These guidelines convert various antidepressant doses into equivalent imipramine levels, allowing for a more accurate comparison of treatments across patients. This approach ensured consistency in data analysis and enhanced the interpretation of clinical characteristics in Table 1. The guidelines are accessible at [http://www.jsprs.org/en/equivalence.tables/] (accessed on 20 February 2023). HCs were recruited from the local community using the Structured Clinical Interview for DSM Disorders (SCID). The inclusion criteria for HCs at baseline were as follows: (1) aged 20 to 64 years at the time of enrollment, with age and gender matched as closely as possible to the MDD group; (2) no diagnosis of depression, other psychiatric disorders, or neurodegenerative conditions; and (3) no history of hypertension, diabetes, dyslipidemia, malignancy, heart failure, renal failure, or other severe comorbidities requiring treatment. Informed consent was obtained from all eligible participants following a comprehensive explanation of this study prior to their participation.

**Laboratory analysis:** Blood samples were obtained from the peripheral vein after 12 h of fasting at the study center. Plasma samples were collected in tubes containing ethylenediamine tetra acetic acid (EDTA) as an anticoagulant and centrifuged at 2000 rpm for 15 min at 4 °C within 30 min of collection. Serum samples were collected in tubes without anticoagulant and allowed to clot at room temperature for 1 h before centrifugation at 2000 rpm for 20 min at 4 °C. The supernatant of both plasma and serum samples was stored at −80 °C until further experiment. Apelin levels were assayed using a commercially available enzyme immunoassay kit (Human Apelin enzyme-linked immunosorbent assay [ELISA] Kit (EEL026), Thermo Fisher Scientific Inc., Carlsbad, CA, USA) following the manufacturer’s instructions. Briefly, both serum and plasma samples (50 μL) and standards were added to a pre-coated plate, followed by incubation with biotinylated antibodies and streptavidin–horseradish peroxidase. After 30 min incubation and further washing, the substrate solution was added and after 15 min incubation stop solution was added. Absorbance was measured at 450 nm. The analytical sensitivity of the assay was 37.5 pg/mL. The standard curve, for the range of 0–4000 pg/mL, was generated by GraphPad Prism 8 (GraphPad Software, La Jolla, CA, USA) using a four-parameter algorithm. The concentration of apelin immunoreactivity in samples was determined by interpolation from the standards curve. Each sample was tested in duplicate.

**Statistical analysis:** All analyses were performed using the Statistical Package for Social Sciences software (SPSS, version 17.0; SPSS Inc., Chicago, IL, USA). The normal distribution of the data was evaluated using skewness and kurtosis values. Continuous variables were presented as the mean ± standard deviation (SD), and categorical variables were presented as percentages. Differences in categorical variables among the groups were examined using the Chi-square test. Continuous variables were compared in the patients with MDD and HC groups using Student’s t-test, while the relationship between apelin levels and MADRS scores was assessed in patients with MDD using partial correlation analysis, with sex, age, BMI, age at depression onset, and imipramine dosage included as covariates. A two-tailed *p* value of <0.05 was considered statistically significant.

## 5. Conclusions

In conclusion, our study contributes to the growing body of research on apelin and highlights its potential relevance as a biomarker for MDD, warranting further investigation. The observed elevation in plasma apelin levels and their significant correlation with the severity of depressive symptoms provide compelling evidence for apelin as one of the molecular targets to consider in mood disorders. Furthermore, these findings support the hypothesis that apelin, a neuropeptide with complex interactions within the neurobiological framework of MDD, may offer valuable insights into MDD pathogenesis. The correlation between apelin levels and depressive symptomatology suggests that further investigation of its neurobiological effects and relationship with the underlying mechanisms of MDD is warranted. Such research could contribute to a more nuanced understanding of MDD and potentially reveal new avenues for exploring the underlying mechanisms of action.

## Figures and Tables

**Figure 1 ijms-25-13663-f001:**
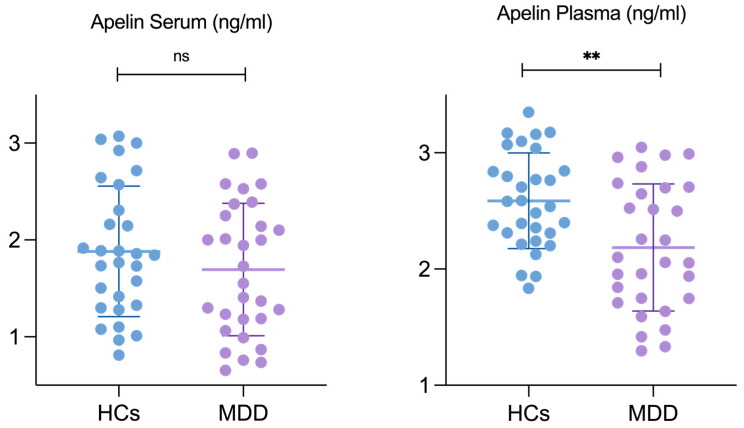
Apelin level differences in serum and plasma between HCs and patients with MDD. Abbreviations: HCs, healthy controls; MDD, patients with major depressive disorder. ** Results are considered statistically significant at *p* < 0.05 level.

**Figure 2 ijms-25-13663-f002:**
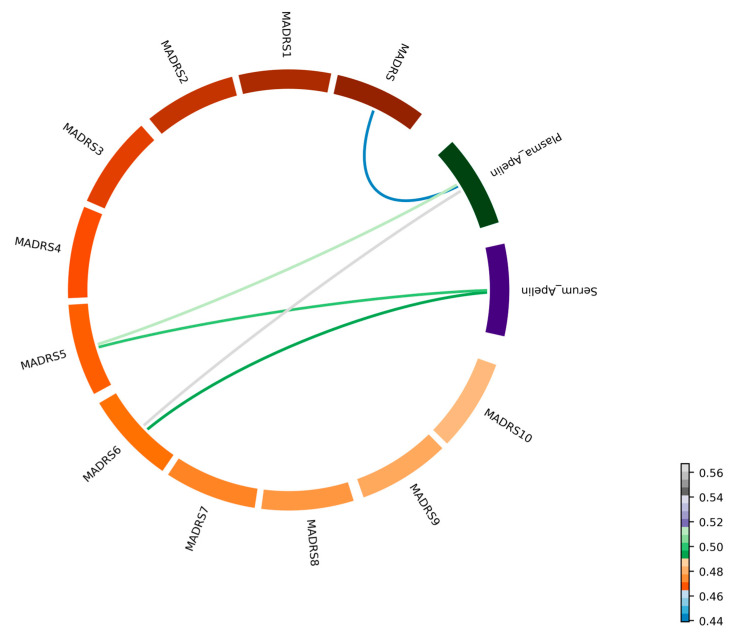
**Correlation between apelin levels and MADRS score (*n* = 30 subjects).** A chord diagram illustrating the partial correlations among the serum and plasma apelin concentrations, the total MADRS score, and the 10 individual MADRS items. Each segment of the circle is labeled with the corresponding variable name, providing clarity on the data represented. Colored lines connect the segments, with the color gradient indicating the strength and direction of the correlations.

**Figure 3 ijms-25-13663-f003:**
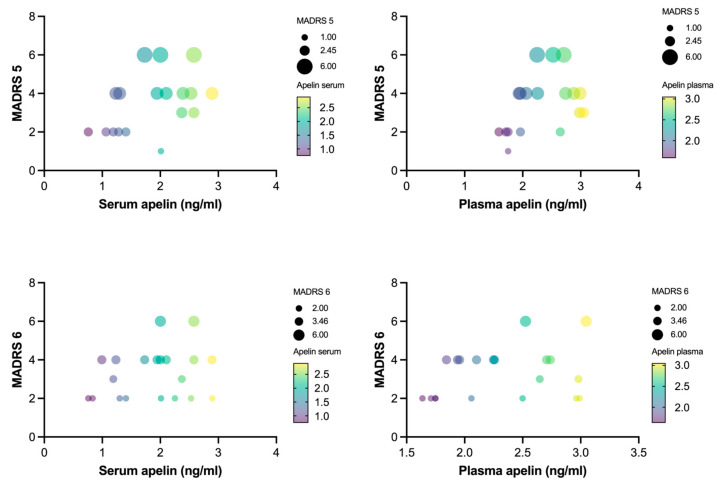
Correlation between apelin levels and MADRS items 5 and 6 (*n* = 30). Partial correlations between apelin concentration in both plasma and serum and Montgomery–Åsberg Depression Rating Scale items 5 and 6. Bubble size represents MADRS score of individual patients with MDD, and gradient color of bubble represents apelin concentration in both serum and plasma.

**Table 1 ijms-25-13663-t001:** Demographic and clinical characteristics of healthy controls and patients with major depressive disorders.

Characteristics	HCs (*n* = 30)	MDD (*n* = 30)
Age (years)	41.47 ± 9.801	44.50 ± 11.717
Gender		
Female (*n*, %)	16 (53.3%)	16 (53.3%)
Body height (cm)	165.36 ± 7.43	161.76 ± 9.13
Body weight (kg)	63.18 ± 9.86	64.01 ± 13.04
BMI (kg/m^2^)	23 ± 2.44	24.24 ± 3.52
Depression onset age (years)	-	39.24 ± 11.36
MADRS	-	22.28 ± 6.36
Imipramine * (mg/day)	-	156.67 ± 92.37

* Imipramine dosage reflects the equivalent dosage of the administered antidepressant drugs in patients, according to national guidelines. Abbreviations: HCs, healthy controls; MDD, patients with major depressive disorder; BMI, body mass index; MADRS, Montgomery–Åsberg Depression Rating Scale.

**Table 2 ijms-25-13663-t002:** Apelin level differences in serum and plasma between healthy controls and patients with major depressive disorder.

	HCs (30)		MDD (30)		*F*	*p*-Value
	Mean	SD	Mean	SD		
Apelin serum (ng/mL)	1.882	0.673	1.694	0.684	1.032	0.293
Apelin plasma (ng/mL)	2.587	0.411	2.185	0.547	1.770	0.002 *

Abbreviations: HCs, healthy controls; MDD, patients with major depressive disorder. * Results are considered statistically significant at *p* < 0.05.

**Table 3 ijms-25-13663-t003:** Correlations between apelin in both plasma and serum and Montgomery–Åsberg Depression Rating Scale scores.

MADRS (*n* = 30)	Apelin Serum (r-Value)	Apelin Plasma (r-Value)
MADRS Total	0.290	0.439 *
MADRS 1 (Apparent Sadness)	0.135	0.229
MADRS 2 (Reported Sadness)	0.215	0.378
MADRS 3 (Inner Tension)	0.009	0.188
MADRS 4 (Reduced Sleep)	0.190	0.246
MADRS 5 (Reduced Appetite)	0.499 *	0.512 *
MADRS 6 (Concentration Difficulties)	0.494 *	0.567 *
MADRS 7 (Lassitude)	0.170	0.274
MADRS 8 (Inability to Feel)	0.014	0.167
MADRS 9 (Pessimistic Thoughts)	0.188	0.242
MADRS 10 (Suicidal Thoughts)	0.143	0.304

Abbreviations: MADRS, Montgomery–Åsberg Depression Rating Scale. In Figure 2, prominent associations between plasma and serum apelin concentrations and MADRS items 5 (reduced appetite) and 6 (reduced sleep) are reported, suggesting potential associations between apelin levels and these specific symptoms. Additionally, Figure 3 presents significant partial correlation results as bubble plot graphs. * Correlation is considered statistically significant at *p* < 0.05.

## Data Availability

No new data were created or analyzed in this study.

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
