# Peer review of "Evaluating Apelin as a Potential Biomarker in Major Depressive Disorder: Its Correlation with Clinical Symptomatology"

_ijms, 2024, doi:10.3390/ijms252413663_

Round 1
Reviewer 1 Report
Comments and Suggestions for Authors
The paper written by Chibaatar et al. entitled: “ Evaluating apelin as a potential biomarker in Major Depressive Disorder: Correlation with clinical symptomatology” presents original results concerning the measurement of total apelin serum and plasma levels in a group of patients with depression, as compared with a group of control subjects, without a history or familiarity of mental illnesses. The study’s concept is of actual interest, falling in the field of translational psychiatry and in the search of peripheral, non-invasive and informative biomarkers of mental disorders; moreover, apelin and related peptide isoforms represent a still scarcely explored molecular pattern in this scientific topic.
However, the study seems to be in a very preliminary state and the results appear ambiguous, difficult to interpret at this stage, suggesting the need for a much more detailed presentation. In its current form, the article is rather difficult to interpret. Suggestions are provided here to improve the manuscript in order to be accepted for publication.
Abstract
Lines 9-10: Please change as follows: “Few and limited studies are available on the potential effects of apelin on mood regulation and emotional behavior”.
Lines 20-22: The conclusive sentence is too emphatic, in respect to the reported results, needing to be modified, as for instance: “This preliminary findings, although not conclusive, suggest that apelin profiles might distinguish patients within MDD, through a deeper evaluation of different apelin isoforms, as well as in different depressed patients”.
Introduction
In the first part of the Introduction, the pathogenetic bases of depression, and the involvement of neuroendocrine axis should be reported too. A short but exhaustive state-of-the art on the search and clinical usefulness of peripheral biomarkers in mental disorders, and particularly depression, is required. Please consider these important aspects. Also, the authors should describe apelin structure, its heterogeneity, its profiles, body distribution, and functions.
Lines 51-55. Please change as suggested, or similarly: “Specifically, the administration of apelin-13 at the intracerebroventricular (ICV) level in rats, was able to reverse depression-like behaviors induced by chronic stress procedures including social defeat or unpredictable chronic mild stress (UCMS)”.
Lines 56-58: This part is quite confusing: try to improve as suggested, or similarly: “Several studies report that apelin exerts antidepressant-like effects by restoring the functionality of the hypothalamus-pituitary-adrenal axis (HPA); however, the exact mechanisms of this action remain unclear, while the data on blood apelin concentrations are still inconsistent in clinical studies”.
Lines 59-65: This sentence should be shifted into the Discussion section. Instead of this sentence, the authors should rather point out, in the Introduction section, why their investigation can provide more information than previous inconsistencies, e.g., explaining the usefulness of investigating total apelin levels in both serum and plasma, of investigating total apelin instead of the various isoforms, and its rationale.
Results
Table 1: Age, Depression age, BMI and Imipramine: the unit of measurement is missing, not specified, please report them.
Lines 86-87: please modify as that: “The mean serum apelin concentration of patients with MDD was 0.734 ± 0.283 pg/ml and 0.668 ± 0.335 pg/ml in HCs”.
Table 3: MADRS 1, MADRS 2, ..etc: please report the symptom cluster denomination, for instance: MADRS 1, apparent sadness; MADRS 2: reported sadness, etc..
Figure 1: The Figure should be presented as a graph showing all single values.
Figure 2: There is no explanation as concerns this figure. Please describe.
The two significant negative Pearson correlations obtained here should be shown in the article as x (MDRS score), y (apelin levels) graphs containing single values. An additional figure is thus required. In fact, if apelin levels were found to be significantly higher in plasma and also higher in serum, albeit not significantly, of MDD patients compared with controls, this would rather imply that apelin levels tend to be increased in MDD. Therefore, it is difficult to understand the reported negative correlation scores. Perhaps, the negative correlations could be due to a small subgroup of patients with more severe MDRS 5 and 6 scores and lower apelin levels, suggesting the possibility of opposite profiles of total apelin levels between different patients. Please report these graphs to provide additional information and give possible interpretations.
Discussion
The discussion needs to be presented in a less confusing way. In addition, several relevant points need to be taken into account. A question arises: is there information on variants of the apelin peptide expressed differently in the two biological matrices, serum and plasma? Then, as indicated in the Method section, some correlations between apelin levels and clinical results are missing, which should instead be taken into account; figures are also missing regarding the observed negative correlations, as already stated, to give a more tangible interpretation of the results in this section. Next: to improve the presentation, authors should maintain a clear line: first, they should discuss their own apelin values compared to those obtained by other authors. Indeed, the levels presented here seem very low, compared, for example, with those presented by Owen et al, Peptides 2021;136:170440. Authors should then discuss first those papers in the previous literature reporting increased levels of apelin in serum or plasma of patients with depression; second, those reporting reduced levels. Authors should also highlight those literature papers reporting total apelin levels, if any, and then those that measure specific variants of the peptide. As it stands, the discussion does not permit interpretations and possible relevance of the preliminary results obtained here.
Materials and Methods
Line 211: Symptoms were assessed using MADRS. Please shortly explain this scale.
Authors should provide more information about patients’ recruitment: drug treatment admission, the choice of patients' drug treatment: imipramine only? Although this is a preliminary investigation, patients could have been better defined by the DSM-5 instrument, indicating the presence of symptoms of other mental conditions as psychosis, anxiety, obsessive-compulsive rituals, atypical features and others. Depression is a highly heterogeneous mental condition and authors should better describe their enrolled patients. Did they were selected on the basis of a similar clinical conditions? There is no inclusion or exclusion criteria in the text. Authors should consider these important points.
Provide information on the statistical test used to depict Figure 2.
Line 224: provide short information about the ELISA procedure and calibration curve calculation; provide the sensitivity of the assay employed.
Another issue: the study’s design should consider correlations between apelin serum and plasma levels and other patients’ information, such as age of depression onset, BMI or Imipramine dosage.
Conclusions
Authors should avoid sentences as that: "In conclusion, our study improves the understanding of apelin as a relevant biomarker for MDD.". This is a too strong statement.
Comments on the Quality of English LanguageEnglish language requires a more accurate revision.
Author Response
Response to Reviewer #1:
The paper written by Chibaatar et al. entitled: “Evaluating apelin as a potential biomarker in Major Depressive Disorder: Correlation with clinical symptomatology” presents original results concerning the measurement of total apelin serum and plasma levels in a group of patients with depression, as compared with a group of control subjects, without a history or familiarity of mental illnesses. The study’s concept is of actual interest, falling in the field of translational psychiatry and in the search of peripheral, non-invasive, and informative biomarkers of mental disorders; moreover, apelin and related peptide isoforms represent a still scarcely explored molecular pattern in this scientific topic.
However, the study seems to be in a very preliminary state and the results appear ambiguous, difficult to interpret at this stage, suggesting the need for a much more detailed presentation. In its current form, the article is rather difficult to interpret. Suggestions are provided here to improve the manuscript in order to be accepted for publication.
We appreciate the time and effort you have dedicated to evaluating our manuscript. Your valuable feedback has been instrumental in enhancing the clarity and strength of our paper. Below, we provide our point-by-point responses to address each comment and suggestion, detailing the revisions and improvements made to the manuscript.
Abstract
Lines 9-10: Please change as follows: “Few and limited studies are available on the potential effects of apelin on mood regulation and emotional behavior”.
Thank you for this suggestion. We have revised the sentence to improve clarity as “To date, only a limited number of studies have investigated the potential effects of apelin on mood regulation and emotional behavior.” on page 1, lines 9-10.
Lines 20-22: The conclusive sentence is too emphatic, in respect to the reported results, needing to be modified, as for instance: “These preliminary findings, although not conclusive, suggest that apelin profiles might distinguish patients within MDD, through a deeper evaluation of different apelin isoforms, as well as in different depressed patients”.
Thank you for this valuable feedback. We agree that the conclusive sentence should more accurately reflect the preliminary nature of our findings. In response, we have revised the conclusion to avoid overstatement and more accurately convey the tentative implications of our study. The revision is added on page 1, lines 20-23 as “These preliminary findings, although not definitive, suggest that apelin profiles may help to identify distinct subgroups within MDD patients, warranting further investigation into different apelin isoforms and their associations in different populations of MDD patients.”.
Introduction
In the first part of the Introduction, the pathogenetic bases of depression, and the involvement of neuroendocrine axis should be reported too. A short but exhaustive state-of-the art on the search and clinical usefulness of peripheral biomarkers in mental disorders, and particularly depression, is required. Please consider these important aspects. Also, the authors should describe apelin structure, its heterogeneity, its profiles, body distribution, and functions.
Thank you for your valuable suggestions. We have revised the Introduction to provide a more comprehensive background and the following additions have been made with additional citations on page 1, lines 29–39 and 43–45.
- Pathogenetic bases of depression and the neuroendocrine axis: Pathologically, it is characterized by a range of brain structural and functional alterations, including neurotransmitter imbalances 3,4, neuroendocrine dysregulation 5,6, neuroinflammation, 7,8 and oxidative stress 9,10. Furthermore, genetic predispositions 11,12 and environmental stressors 13 significantly affect these pathological changes. In addition, hypothalamus-pituitary-adrenal (HPA) axis dysregulation is a critical component of the pathophysiology of MDD, as it mediates the stress response and may perpetuate depressive symptoms through sustained neuroendocrine disruption 14.
- Peripheral biomarkers in depression: In recent years, there has been growing interest in identifying peripheral biomarkers that could enhance our understanding of the pathophysiology of MDD and facilitate accurate diagnosis, rapid treatment initiation, and effective disease monitoring 15.
- Apelin structure, heterogeneity, disruption, and function: Apelin, an endogenous peptide first isolated in 1998 from the bovine stomach by Fujino et al. 16, has emerged as an important neuropeptide in animals and humans. Its biological action is mediated by the high-affinity single G protein-coupled apelin receptor (APJ). Apelin has multiple endogenous bioactive isoforms ranging from 13 to 55 amino acids exerting a variety of important functions in both physiological and pathological conditions 17. Apelin and APJ are extensively expressed in several peripheral organs, including the heart, kidneys, lungs, and stomach 18–23 where they positively influence important physiological processes such as cardiovascular regulation, fluid homeostasis, energy metabolism, and neuroendocrine function 24.
Lines 51-55. Please change as suggested, or similarly: “Specifically, the administration of apelin-13 at the intracerebroventricular (ICV) level in rats, was able to reverse depression-like behaviors induced by chronic stress procedures including social defeat or unpredictable chronic mild stress (UCMS)”.
Thank you for your suggestion. We have modified the sentence accordingly as “Specifically, intracerebroventricular (ICV) administration of apelin-13 in rats was shown to reverse depression-like behaviors induced by chronic stress models, including social defeat or unpredictable chronic mild stress (UCMS).” on page 2, lines 59–62.
Lines 56-58: This part is quite confusing: try to improve as suggested, or similarly: “Several studies report that apelin exerts antidepressant-like effects by restoring the functionality of the hypothalamus-pituitary-adrenal axis (HPA); however, the exact mechanisms of this action remain unclear, while the data on blood apelin concentrations are still inconsistent in clinical studies”.
Thank you very much for the comment to improve clarity and flow. We have revised the section as “Numerous studies have suggested that apelin may exert antidepressant-like effects, potentially through the modulation of the hypothalamus-pituitary-adrenal (HPA) axis 34,35. However, the precise mechanisms underlying these effects remain poorly understood, and findings regarding apelin concentrations in blood remain inconsistent across clinical studies.” on page 2, lines 63–67.
Lines 59-65: This sentence should be shifted into the Discussion section. Instead of this sentence, the authors should rather point out, in the Introduction section, why their investigation can provide more information than previous inconsistencies, e.g., explaining the usefulness of investigating total apelin levels in both serum and plasma, of investigating total apelin instead of the various isoforms, and its rationale.
Thank you for your insightful comment. We have moved the mentioned section into the Discussion section to enhance the clarity of the manuscripts. Moreover, we fully agree with your suggestion to elaborate on the usefulness of our investigation. To address this, we have added following section “Given the limited understanding of the roles of specific isoforms, assessing total apelin in both serum and plasma may offer a useful measure of the systemic concentration in patients with MDD. Also, it could allow for cross-validation of findings and facilitate analysis by representing the combined effect of all bioactive isoforms on apelin receptors.” and “This aspect of the study might reveal preliminary associations in MDD and allow to explore the potential predictive value of total apelin levels for depression severity, thereby providing insights that could guide into the potential of apelin as a biomarker in clinical settings.” on page 2, lines 70–74 and 77–80.
Results
Table 1: Age, Depression age, BMI and Imipramine: the unit of measurement is missing, not specified, please report them.
Thank you very much for bringing this to our attention. We have carefully addressed the issue and added the unit of measurement in Table 1 and Results section on page 2.
Lines 86-87: please modify as that: “The mean serum apelin concentration of patients with MDD was 1.694 0.684 ng/ml and 1.882 0.673 ng/ml in HCs”.
We appreciate your careful review which has helped improve the clarity and accuracy of our manuscript. We have made the change as you suggested on page 3, line 95–96.
Table 3: MADRS 1, MADRS 2, ..etc: please report the symptom cluster denomination, for instance: MADRS 1, apparent sadness; MADRS 2: reported sadness, etc..
Thank you for highlighting this important point. We have revised Table 3 and added all the symptom cluster denomination for each MADRS.
Figure 1: The Figure should be presented as a graph showing all single values.
Thank you for your suggestion. We have revised the figure to present it as a graph displaying all individual data points, as recommended. It can ensure a clearer representation of the variability and distribution within the dataset.
Figure 2: There is no explanation as concerns this figure. Please describe.
Thank you for pointing this out. We have revised the manuscript to include more detailed information about the figure as “A chord diagram illustrating the correlations between serum and plasma apelin concentrations, the total MADRS score, and the 10 individual MADRS items. Each segment of the circle is labeled with the corresponding variable name, providing clarity on the data represented. Colored lines connect the segments, with the color gradient indicating the strength and direction of the correlations. Notably, prominent connections are observed between plasma and serum apelin concentrations and MADRS items 5 (reduced appetite) and 6 (reduced sleep), suggesting potential associations between apelin levels and these specific symptoms. Additionally, a weak negative correlation is evident between plasma apelin concentration and total MADRS score.” on page 5.
The two significant negative Pearson correlations obtained here should be shown in the article as x (MDRS score), y (apelin levels) graphs containing single values. An additional figure is thus required. In fact, if apelin levels were found to be significantly lower in plasma and also lower in serum, albeit not significantly, of MDD patients compared with controls, this would rather imply that apelin levels tend to be increased in MDD. Therefore, it is difficult to understand the reported negative correlation scores. Perhaps, the negative correlations could be due to a small subgroup of patients with more severe MDRS 5 and 6 scores and lower apelin levels, suggesting the possibility of opposite profiles of total apelin levels between different patients. Please report these graphs to provide additional information and give possible interpretations.
We are appreciated for your detailed feedback and thoughtful observation. We have taken your comments into account and revised the manuscript accordingly:
- Additional figures: We have included bubble plots to illustrate significant Pearson correlations. In this plot:
- x-axis: apelin concentration
- y-axis: MADRS score
- bubbles are representing individual patients with different size which represents MADRS score and gradient color which represents apelin concentration levels.
- Regarding the potential influence of a small subgroup of patients with severe MADRS 5 and 6 scores, we calculated the average scores for all MADRS items, and the results are as follows.
|
MADRS 1 |
MADRS 2 |
MADRS 3 |
MADRS 4 |
MADRS 5 |
MADRS 6 |
MADRS 7 |
MADRS 8 |
MADRS 9 |
MADRS 10 |
|
1.77 |
2.67 |
2.4 |
2.3 |
2.1 |
2.2 |
2 |
2.2 |
2.5 |
1.9 |
As shown, the average scores for MADRS 5 and 6 are not notably higher than other items, suggesting that the negative correlations are unlikely to be driven by severe MADRS 5 and 6 scores.
Discussion
The discussion needs to be presented in a less confusing way. In addition, several relevant points need to be taken into account.
Thank you for your constructive and detailed feedback. Below, we provide our point-by-point responses to address each comment and suggestion, detailing the revisions and improvements made to the manuscript.
A question arises: is there information on variants of the apelin peptide expressed differently in the two biological matrices, serum, and plasma?
Currently, there is limited information on the differential expression of apelin variants between serum and plasma, particularly in patients with MDD. This knowledge gap was one of the primary motivations for initiating our research, as we sought to investigate whether such differences exist. Furthermore, we anticipated that measuring total apelin levels in both serum and plasma could provide a more comprehensive understanding and potentially simplify future research efforts, facilitating the adaptation of apelin measurement into clinical assessments.
Then, as indicated in the Method section, some correlations between apelin levels and clinical results are missing, which should instead be taken into account;
Thank you for raising this point. Correlation analyses were conducted between apelin levels (in both serum and plasma) and various patient-related variables, including age of depression onset, BMI, and imipramine dosage. However, no significant correlations were observed between these variables in our dataset.
figures are also missing regarding the observed correlations, as already stated, to give a more tangible interpretation of the results in this section.
We have added the related graphs with individual variables to visualize the correlations.
Indeed, the levels presented here seem very low, compared, for example, with those presented by Owen et al, Peptides 2021;136:170440.
Thank you very much for bringing this to our attention. We acknowledge an oversight in presenting the results using optical density instead of apelin concentrations. Upon careful review, we have corrected this mistake and reanalyzed the data, reporting the results in the appropriate unit of ng/ml for clarity. Revisions have been made throughout all relevant sections of the manuscript to ensure the results are presented in a more accurate and comprehensible manner.
Next: to improve the presentation, authors should maintain a clear line: first, they should discuss their own apelin values compared to those obtained by other authors. Authors should then discuss first those papers in the previous literature reporting reduced levels of apelin in serum or plasma of patients with depression; second, those reporting increased levels. Authors should also highlight those literature papers reporting total apelin levels, if any, and then those that measure specific variants of the peptide. As it stands, the discussion does not permit interpretations and possible relevance of the preliminary results obtained here.
Thank you for your valuable feedback. To improve the clarity of the discussion, we have reorganized the section to follow a more structured approach. All related changes have been highlighted in blue for your convenience.
Materials and Methods
Line 211: Symptoms were assessed using MADRS. Please shortly explain this scale.
Thank you for pointing this out. We have revised the manuscript include a brief explanation of MADRS with additional citation as “Symptoms were assessed using MADRS, that commonly used assessment method to evaluate depression which can be administered relatively quickly and more sensitive to address core symptoms such as sadness, tension, lassitude, pessimistic thoughts, and suicidal thoughts 50,51.” on page 7, lines 240–257.
Authors should provide more information about patients’ recruitment: drug treatment admission, the choice of patients' drug treatment: imipramine only? Although this is a preliminary investigation, patients could have been better defined by the DSM-5 instrument, indicating the presence of symptoms of other mental conditions as psychosis, anxiety, obsessive-compulsive rituals, atypical features, and others. Depression is a highly heterogeneous mental condition and authors should better describe their enrolled patients. Did they were selected on the basis of a similar clinical conditions? There is no inclusion or exclusion criteria in the text. Authors should consider these important points.
We recognize the importance of providing comprehensive information about patient recruitment and selection criteria. To address these concerns, we have revised the manuscript as follows: “Patients included in the study met the following criteria at baseline: (1) aged 20 to 64 years at the time of enrollment, regardless of gender; (2) diagnosed with major depressive disorder (MDD) according to DSM-5 criteria, with a duration of symptoms between 4 and 12 weeks; (3) no history or comorbidity of neurodegenerative disorders (e.g., dementia) or other psychiatric conditions aside from MDD; and (4) absence of serious comorbidities requiring inpatient treatment. There were no restrictions regarding psychotropic medication use or the frequency of hospital visits for outpatient participants. Informed consent was obtained from all eligible patients prior to their participation in the study. HCs were recruited from the local community using the Structured Clinical Interview for DSM Disorders (SCID). The inclusion criteria for HCs at baseline were as follows: (1) aged 20 to 64 years at the time of enrollment, with age and gender matched as closely as possible to the MDD group; (2) no diagnosis of depression, other psychiatric disorders, or neurodegenerative conditions; and (3) no history of hypertension, diabetes, dyslipidemia, malignancy, heart failure, renal failure, or other severe comorbidities requiring treatment. Informed consent was obtained from all eligible participants following a comprehensive explanation of the study prior to their participation.” on page 8, lines 243–257.
Line 224: provide short information about the ELISA procedure and calibration curve calculation; provide the sensitivity of the assay employed.
Thank you for your comment. The following brief information has been added to clarify ELISA procedure: “Briefly, both serum and plasma samples (50 μl) and standards were added to a pre-coated plate, followed by incubation with biotinylated antibodies and streptavidin-horseradish peroxidase. After 30 minutes incubation and further washing, the substrate solution was added and after 15 minutes incubation stop solution was added. Absorbance was measured at 450 nm. The analytical sensitivity of the assay is 37.5 pg/ml. The standard curve, the range of 0-4000pg/ml, is generated by GraphPad Prism 8 (GraphPad Software, La Jolla, CA, USA) using a four-parameter algorithm. The concentration of apelin immunoreactivity in samples was determined by interpolation from the standards curve. Each sample was tested in duplicates.” on page 8, lines 267–275 in Laboratory Analysis section.
Another issue: the study’s design should consider correlations between apelin serum and plasma levels and other patients’ information, such as age of depression onset, BMI or Imipramine dosage.
Thank you for raising this point. Correlation analyses were conducted between apelin levels (in both serum and plasma) and various patient-related variables, including age of depression onset, BMI, and imipramine dosage. However, no significant correlations were observed between these variables in our dataset.
Conclusions
Authors should avoid sentences as that: "In conclusion, our study improves the understanding of apelin as a relevant biomarker for MDD.". This is a too strong statement.
Thank you for this feedback. We agree that the statement could be perceived as overly definitive. To address this, we have revised the conclusion as “In conclusion, our study contributes to the growing body of research on apelin and highlights its potential relevance as a biomarker for MDD, warranting further investigation.” to convey a more balanced interpretation of our findings.

Reviewer 2 Report
Comments and Suggestions for Authors
The current study explores an interesting topic with potentially useful clinical implications, especially given the acute lack of biomarkers in Psychiatry. There are some methodological aspects that should be clarified, in addition to several formal details.
Line 28- Please insert a comma between 25 and 26;
Table 1- Why is only imipramine mentioned in this table? Does this mean all patients were on imipramine? What is the unit of measure for the values indicated for imipramine, i.e., mg/day, or plasma concentrations (what units)? What does „depression age” mean, i.e., the length of depression, or teh age of the patients at the MDD’s onset?
Lines 106-110- Please specify which are items 5 and 6 of the MADRS; I saw the definitions are mentioned in lines 129-130, but it would be useful for the readers to see these definitions here, too;
Line 105- „was statistically significant” or „reached the level of statistical significance”;
Line 111- Please use „r” for consistency instead of „R”;
Line 169-170- Please change the variables into „MADRS5 and MADRS6” for consistency;
Lines 202-233- maybe consider inserting this chapter before the „Results” section; lines 205-207- please insert the number and date of approval from the Local Ethics Committee; line 211- please replace „DSM-V” with „DSM-5”. How were the patients selected (e.g., randomly, convenience sample, all patients evaluated during a pre-determined period)? Were healthy controls screened for psychiatric disorders? Please detail the inclusion and exclusion criteria because confounders may exist in both groups, like metabolic disorders, neurological diseases, cardiovascular diseases, etc. (e.g., https://pmc.ncbi.nlm.nih.gov/articles/PMC9464848/, https://pubmed.ncbi.nlm.nih.gov/38089606/).
Ref. 47- The complete citation for this source is:
American Psychiatric Association. Diagnostic and statistical manual of mental disorders, 5th edition. Arlington, American Psychiatric Publishing, 2013.
Author Response
Response to Reviewer #2:
The current study explores an interesting topic with potentially useful clinical implications, especially given the acute lack of biomarkers in Psychiatry. There are some methodological aspects that should be clarified, in addition to several formal details.
We appreciate all the time and effort that you committed to strengthening our paper and our point-by-point responses as follows:
Table 1- Why is only imipramine mentioned in this table? Does this mean all patients were on imipramine? What is the unit of measure for the values indicated for imipramine, i.e., mg/day, or plasma concentrations (what units)? What does „depression age” mean, i.e., the length of depression, or the age of the patients at the MDD’s onset?
We are apologizing for the misunderstanding caused by not presenting the variables in sufficient details. While not all patients were treated with imipramine specifically, the dose of each patient’s antidepressant was converted to imipramine equivalents for consistency and comparability. The imipramine equivalent scale, widely utilized in Japan, serves as a standard for dose equivalency across various antidepressants. This approach allows for the assessment of antidepressant treatment effects in a uniform manner, facilitating comparison between different therapeutic regimens. Further details on the imipramine equivalent scale can be found in the reference provided (http://www.jsprs.org/en/equivalence.tables/). Moreover, we have revised the Table 1 and added measuring unit of mg/day. As for depression age it’s the age of the patients at the MDD’s onset and we have changed “Depression age” into “Depression onset age” to provide clarity.
Lines 106-110- Please specify which are items 5 and 6 of the MADRS; I saw the definitions are mentioned in lines 129-130, but it would be useful for the readers to see these definitions here, too;
Thank you very much for pointing this out. Your comment has been considered as we added “MADRS 5 (reduced appetite) and MADRS 6 (concentration difficulties)” wherever these aspects are referenced.
Line 105- „was statistically significant” or „reached the level of statistical significance”;
Thank you for your insightful feedback. We have made suggested revision, updating "was statistically significance" to "was statistically significant".
Line 111- Please use „r” for consistency instead of „R”;
Thank you for your precise observation. We have revised "R-values" to "r-values".
Line 169-170- Please change the variables into „MADRS5 and MADRS6” for consistency;
Thank you for your valuable observation. We have revised "MAD5 and MAD6" as "MADRS 5 and MADRS 6" on page7, line 197.
Lines 202-233- maybe consider inserting this chapter before the „Results” section;
We sincerely appreciate your thoughtful suggestion. However, due to the requirements of the IJMS template, we regret that we are unable to modify the chapter placement.
lines 205-207- please insert the number and date of approval from the Local Ethics Committee;
We are thankful for your recommendation and we have added approval number and date as “approval code: UOEHCRB21-164, approval date: 2022/01/19” on page 7, line 235.
line 211- please replace „DSM-V” with „DSM-5”. How were the patients selected (e.g., randomly, convenience sample, all patients evaluated during a pre-determined period)? Were healthy controls screened for psychiatric disorders? Please detail the inclusion and exclusion criteria because confounders may exist in both groups, like metabolic disorders, neurological diseases, cardiovascular diseases, etc. (e.g., https://pmc.ncbi.nlm.nih.gov/articles/PMC9464848/, https://pubmed.ncbi.nlm.nih.gov/38089606/).
Thank you for your suggestion, We have replaced DSM-V with DSM-5. Also, we recognize the importance of providing comprehensive information about patient recruitment and selection criteria. To address these concerns, we have revised the manuscript as follows: “Patients included in the study met the following criteria at baseline: (1) aged 20 to 64 years at the time of enrollment, regardless of gender; (2) diagnosed with major depressive disorder (MDD) according to DSM-5 criteria, with a duration of symptoms between 4 and 12 weeks; (3) no history or comorbidity of neurodegenerative disorders (e.g., dementia) or other psychiatric conditions aside from MDD; and (4) absence of serious comorbidities requiring inpatient treatment. There were no restrictions regarding psychotropic medication use or the frequency of hospital visits for outpatient participants. Informed consent was obtained from all eligible patients prior to their participation in the study. HCs were recruited from the local community using the Structured Clinical Interview for DSM Disorders (SCID). The inclusion criteria for HCs at baseline were as follows: (1) aged 20 to 64 years at the time of enrollment, with age and gender matched as closely as possible to the MDD group; (2) no diagnosis of depression, other psychiatric disorders, or neurodegenerative conditions; and (3) no history of hypertension, diabetes, dyslipidemia, malignancy, heart failure, renal failure, or other severe comorbidities requiring treatment. Informed consent was obtained from all eligible participants following a comprehensive explanation of the study prior to their participation.” on page 8, lines 243–257.
Ref. 47- The complete citation for this source is:
American Psychiatric Association. Diagnostic and statistical manual of mental disorders, 5th edition. Arlington, American Psychiatric Publishing, 2013.
Thank you very much for providing the complete citation and we have changed the citation as you suggested on page 10, lines 437–438.

Round 2
Reviewer 1 Report
Comments and Suggestions for Authors
General comments:
The new version of the paper written by Chibaatar et al. entitled: “ Evaluating apelin as a potential biomarker in Major Depressive Disorder: Correlation with clinical symptomatology shows overall improvement, but data, results and methods still present relevant uncertainty. Indeed, results are now opposite than those reported in the previous version of this work: plasma apelin amounts are now significantly decreased in MDD in respect to healthy controls, while in the past version they were increased in patients. This is quite surprising and very difficult to be conceived by a reviewer. How come this new result? Then, the unit of measurement reported in the first version as pg/ml, has been currently changed into ng/ml (see ELISA kit STD calibration range 62.5-4000 pg/ml, Thermofisher Scientific), while values appearing quite different (see Table 2) from the previous ones obtained in the same individuals (see Table 1). Where did the previous data come from? Can you explain this? All these issues indicate data inaccuracy, making results' interpretation very difficult. The paper is not acceptable for publication in these terms, although the topic per se is of scientific and clinical interest.
Specific comments:
Results: Beside what reported in the above General comments: correlations between MADRS items and apelin levels are ambiguous. The authors states that there are negative correlations between apelin levels and MADRS 6 scores and they present positive Pearson coefficient of correlation: “Similarly, MADRS 6 (concentration difficulties) scores showed significant negative (?) correlations with serum ( r = 0.494, p < 0.05) and plasma (r = 0.567, p < 0.05)”, lines 115-117, page 4. Without data clarity there is no coherence in data interpretation. And in the past version these values were effectively reported as negative. Figure 3 would be clearer with typical Pearson scatterplot correlations.
More, in your reply to the point: “The two significant negative Pearson correlations obtained here should be..” the authors shows a table with MADRS average scores per item and write: “ As shown, the average scores for MADRS 5 and 6 are not notably higher than other items, suggesting that the negative correlations are unlikely to be driven by severe MADRS 5 and 6 scores.” So, correlations were positive or negative? There is a problem with the sign of the r value . There is no coherence. Moreover, perhaps I was misunderstood: by more severe symptoms I did not mean more severe in absolute terms, but simply that the lower levels of apelin within patients could correspond to the higher MDRS 6 values, if the correlation was negative, as in the past version. But now it is difficult to understand how these correlations are, and this is really confusing for a reader. I suppose they are all positive, but, I repeat, what is real, the previous negative one or the current positive one? This confusion is not acceptable.
Discussion:
If now patients have average lower levels of total apelin than controls, positive correlation with MDRS scores suggest that patients have relatively higher total apelin levels when they have current more severe symptoms, in particular at MDRS 5, thus greater appetite reduction, and MADRS 6, thus greater concentration difficulties. The authors should better explain this apelin trajectory within patients, if this is indeed the case.
A list of study’s limitations must be provided.
Methods:
There is still a main issue: the authors state that they did not restrict patients'selection on the basis of administered drugs: MDD is a heterogeneous disorder and the study reports in Table 1 that they were all taking Imipramine, a tricyclic antidepressant which is not a first choice compound in depression treatment. This seems quite surprising and atypical.
Comments on the Quality of English Language
English is quite correct, needing some revision.
Author Response
Response to Reviewer #1:
General comments:
The new version of the paper written by Chibaatar et al. entitled: “ Evaluating apelin as a potential biomarker in Major Depressive Disorder: Correlation with clinical symptomatology shows overall improvement, but data, results and methods still present relevant uncertainty. Indeed, results are now opposite than those reported in the previous version of this work: plasma apelin amounts are now significantly decreased in MDD in respect to healthy controls, while in the past version they were increased in patients. This is quite surprising and very difficult to be conceived by a reviewer. How come this new result? Then, the unit of measurement reported in the first version as pg/ml, has been currently changed into ng/ml (see ELISA kit STD calibration range 62.5-4000 pg/ml, Thermofisher Scientific), while values appearing quite different (see Table 2) from the previous ones obtained in the same individuals (see Table 1). Where did the previous data come from? Can you explain this? All these issues indicate data inaccuracy, making results' interpretation very difficult. The paper is not acceptable for publication in these terms, although the topic per se is of scientific and clinical interest.
Dear Reviewer,
We sincerely appreciate your thorough review and valuable feedback, which have been instrumental in improving our manuscript. We acknowledge the critical points you raised regarding the discrepancies in the apelin data and unit of measurement between the initial and revised versions of our manuscript.
Upon careful re-examination of our dataset, we identified a significant oversight in the first version of the manuscript. Specifically, the results were inadvertently presented as Optical Density (OD) values rather than concentrations of both serum and plasma apelin. As you rightly noted in the first-round review as “Indeed, the levels presented here seem very low, compared, for example, with those presented by Owen et al, Peptides 2021;136:170440.”. This prompted us to revisit and reanalyze the raw data. We found that the apelin concentrations had been misinterpreted due to the direct use of OD readings instead of converting them into corresponding concentrations using the standard curve provided by the ELISA kit. Since OD is inversely proportional to concentration, correcting this error resulted in the observed reversal of the apelin level trends, with concentrations being lower in MDD patients compared to HCs, as expected from the corrected analysis. We deeply regret any confusion caused by this oversight and assure you that we have now thoroughly validated the dataset and verified the integrity of the revised results. We have included this explanation in the first-round review and revised the manuscript to prevent any further ambiguity.
Regarding the unit of measurement, we initially reported apelin concentrations in pg/ml. For clarity and ease of interpretation, we converted the values to ng/ml in the revised manuscript (e.g., 1694 ± 684 pg/ml is now presented as 1.694 ± 0.684 ng/ml). This conversion remains within the calibration range of the 62.5–4000 pg/ml.
Once again, we apologize for the oversight and thank you for your meticulous review, which has significantly contributed to enhancing the quality and accuracy of our work. For full transparency, we have included the standard curve graph and data below for your review. We are also happy to provide the complete dataset or any additional information upon your request.
Specific comments:
Results: Beside what reported in the above General comments: correlations between MADRS items and apelin levels are ambiguous. The authors states that there are negative correlations between apelin levels and MADRS 6 scores and they present positive Pearson coefficient of correlation: “Similarly, MADRS 6 (concentration difficulties) scores showed significant negative (?) correlations with serum ( r = 0.494, p < 0.05) and plasma (r = 0.567, p < 0.05)”, lines 115-117, page 4. Without data clarity there is no coherence in data interpretation. And in the past version these values were effectively reported as negative. Figure 3 would be clearer with typical Pearson scatterplot correlations.
Thank you for your insightful comments and for highlighting the ambiguity in the reported correlations between apelin levels and MADRS items. We appreciate your careful review and the opportunity to clarify and improve the accuracy of our data and its presentation.
You are correct in noting the discrepancy between the correlation direction reported in the revised manuscript and the previous version. This change is a direct consequence of our correction of a critical error in the data presentation. In the first version of the manuscript, we inadvertently presented the results using OD values instead of the apelin concentrations derived from the ELISA standard curve. Since OD is inversely proportional to concentration, this resulted in the correlation between MADRS items and apelin levels being incorrectly reported as negative. After reanalyzing the data using the correct apelin concentrations, we observed a positive correlations. We have updated the manuscript to reflect these corrected correlations accurately.
To further improve clarity and transparency, we have added below the figure with Pearson scatterplots illustrating the positive correlation between apelin concentrations both in plasma and MADRS item 5 and 6 scores.
More, in your reply to the point: “The two significant negative Pearson correlations obtained here should be..” the authors shows a table with MADRS average scores per item and write: “ As shown, the average scores for MADRS 5 and 6 are not notably higher than other items, suggesting that the negative correlations are unlikely to be driven by severe MADRS 5 and 6 scores.” So, correlations were positive or negative? There is a problem with the sign of the r value . There is no coherence. Moreover, perhaps I was misunderstood by more severe symptoms I did not mean more severe in absolute terms, but simply that the lower levels of apelin within patients could correspond to the higher MDRS 6 values, if the correlation was negative, as in the past version. But now it is difficult to understand how these correlations are, and this is really confusing for a reader. I suppose they are all positive, but I repeat, what is real, the previous negative one or the current positive one? This confusion is not acceptable.
We sincerely apologize for any confusion caused by the typographical error and previous inconsistencies in our responses. To clarify, all correlations between apelin levels and MADRS item scores, including MADRS 5 (reduced appetite) and MADRS 6 (concentration difficulties), are positive in the revised analysis.
Once again, we sincerely apologize for the confusion caused by the initial error and subsequent inconsistencies in reporting the correlations. The positive correlations presented in the revised manuscript reflect the corrected and validated data. We are confident that these updates resolve the issues and ensure the data’s coherence and reliability. We are deeply grateful for your detailed feedback, which has significantly contributed to improving the clarity and scientific rigor of our manuscript.
Discussion:
If now patients have average lower levels of total apelin than controls, positive correlation with MDRS scores suggest that patients have relatively higher total apelin levels when they have current more severe symptoms, in particular at MDRS 5, thus greater appetite reduction, and MADRS 6, thus greater concentration difficulties. The authors should better explain this apelin trajectory within patients, if this is indeed the case.
Thank you for your insightful comment and for highlighting the need to better explain the relationship between apelin levels and symptom severity, particularly in relation to MADRS 5 (reduced appetite) and MADRS 6 (concentration difficulties).
In our study, we observed that patients with MDD exhibit lower total apelin levels compared to HCs. However, within the MDD group, a positive correlation was found between apelin levels and symptom severity, particularly with MADRS 5 and MADRS 6. A potential explanation for the positive correlation between apelin and MADRS 5 (reduced appetite) is the well-established role of apelin in appetite regulation. In general, higher apelin levels are associated with greater appetite. Therefore, in MDD patients, who already have lower baseline apelin levels, a further reduction in apelin may contribute to greater appetite loss. Conversely, patients with relatively higher apelin levels within the MDD group may experience less severe appetite reduction, which results in a positive correlation.
Similarly, the positive correlation with MADRS 6 (concentration difficulties) may suggest that apelin plays a role in cognitive function and that fluctuations in apelin levels could influence concentration and attention. However, this relationship warrants further investigation to better understand the underlying mechanisms and their potential relevance in the pathophysiology of MDD.
Once again, we are greatly appreciated for your valuable time and efforts in reviewing our work.
Methods:
There is still a main issue: the authors state that they did not restrict patients' selection on the basis of administered drugs: MDD is a heterogeneous disorder and the study reports in Table 1 that they were all taking Imipramine, a tricyclic antidepressant which is not a first choice compound in depression treatment. This seems quite surprising and atypical.
We are apologizing for the misunderstanding caused by not presenting the variables in sufficient details. While not all patients were treated with imipramine specifically, the dose of each patient’s antidepressant was converted to imipramine equivalents for consistency and comparability. The imipramine equivalent scale, widely utilized in Japan, serves as a standard for dose equivalency across various antidepressants. This approach allows for the assessment of antidepressant treatment effects in a uniform manner, facilitating comparison between different therapeutic regimens. Further details on the imipramine equivalent scale can be found in the reference provided (http://www.jsprs.org/en/equivalence.tables/).
Reviewer 2 Report
Comments and Suggestions for Authors
The quality of the manuscript significantly improved
Author Response
Dear Reviewer,
We would like to sincerely thank you for taking the time to review my and greatly appreciate your positive feedback. Your support and consideration are invaluable, and I am pleased that the work has met your expectations. Should you have any further suggestions or inquiries, please do not hesitate to reach out.
Once again, thank you for your time and for contributing to the successful progression of this manuscript.
Round 3
Reviewer 1 Report
Comments and Suggestions for Authors
The latest version of the manuscript of Chibaatar et al, entitled :"Evaluating apelin as a potential biomarker in Major Depressive Disorder: Correlation with clinical symptomatology" has been revised as suggested, and is now much more comprehensible. By this new and clearer version of the paper, it has been possible to check the presence of other issues. Indeed some main methodological issues now emerge, still requiring explanation in order to ensure a correct interpretation of results. Also, additional analyses are recommended, if these have not been carried out, since the manuscript is not informative enough in this respect.
Main points:
Material and methods
This section still requires careful revision and extension.
1) First, the adopted MADRS functioning must be clearly explained in order to be easily understood also by readers who are not directly involved in clinical psychiatry assessments, enabling a full comprehension of the statistical significance reported.
Specifically:
Page 8, lines 239-242: The authors should provide at this point the extensive explanation of the functioning of the administered MADRS scale. This is necessary to interpret correlation results. Indeed, following the standard uses of MADRS questionnaire, the positive Pearson correlations obtained in this work rather reflect that patients showing highest apelin plasma or serum levels also reported the highest scores (higher severity) at the MADRS total questionnaire, as well as at the MADRS 5 and 6 items. Moreover: did the authors interview also non depressed control subjects by the MADRS scale and analyzed their scores in correlations? in Table 1, results seem to indicate that they did not interviewed controls. The authors should perform this evaluation if they have collected these data.
2) Drug treatment - Page 8, line 248: The authors should explain at this point that, in order to standardize patients in respect to the different treatments applied, they used Japanese guidelines for antidepressant drug equivalence, providing the dedicated link. This is important to understand Table 1 and the patients' clinical picture.
Others points and detailed comments:
Introduction
Page 2, Line 47: The authors should change "Furthermore,.." into: " In particular, .."
Page 2, lines 52-55 : " Significant reductions in neurological deficits and infarct volume, and blood-brain barrier (BBB) protection in rat and mouse models of cerebral ischemia were observed 27,28. In addition, it reduced brain damage in mice with traumatic brain injury 29." The sentence is unclear, it should be changed as, for instance: " Apelin has been reported to significantly prevent from and/or reduce neurological deficits, infarct volume, and blood-brain barrier (BBB) damage in rodent models of cerebral ischemia27,28. In addition, this peptide has been found to restrain neuronal damage in mice with traumatic brain injury29."
Page 2, lines 57-58: The authors should rather write: "The potential effects of apelin on mood regulation and emotional behavior have been investigated, but studies conducted so far are limited".
Page 2, lines 68-69: "First, we sought to investigate the differential levels of apelin in serum and plasma of healthy controls (HCs) and patients with MDD." Comment: the authors should clearly and briefly state after this sentence why they investigated these two biological samples, explaining potential differences in apelin concentration in serum and plasma, if experimental and/or physiological, to better introduce the work. Indeed, usually serum and plasma molecular components are at similar concentrations, except for specific compounds participating in clotting processes and platelet release.
Page 2, lines 69-71: The authors should rather modify as indicated here: "Given the poor understanding of the roles of specific isoforms by now, assessing total apelin in both serum and plasma may offer a preliminary useful measure of the systemic concentration in patients with MDD..."
Page 2, lines 77-79: At this point, the authors should change the manuscript as follows: "..allow to explore the potential predictive value of total apelin levels in respect to the presence of depressive symptoms as well as to their degree of severity, thereby providing insights that could guide towards considering this peptide as a biomarker in clinical settings."
Results
Page 3, line 91: The authors should indicate in the Table caption that Imipramine dosage reflects the equivalent dosage of the administered antidepressant drugs in patients, accordingly to national guidelines.
Page 3, lines 97-98: "..with a p-value of 0.002, indicating statistical significance. (Table 2 and Figure 1)". The authors should modify the text as such:"..with a p-value of 0.002, indicating statistical significance. (Table 2). Figure 1 depicts instead the same data reported in Table 2, as a scatterplot graphical representation of single values". The authors should also add a short sentence indicating that their patients resulted moderately depressed from Table 1 results.
Page 3, lines 105-106: The authors should modify the text such as: "MDD, patients with major depressive disorder".
Page 3, line 109: The authors should report this sentence as: "Detailed Pearson correlation data are provided in Table 3 and Figure 2".
Page 4, Table 3: The Table presentation is not clear as that. The authors should name columns as: MADRS, Apelin serum, r coefficient, Apelin plasma, r coefficient.
Page 4, line 123: The authors should provide a brief presentation of results presented in Figure 2 and figure 3. For instance, lines 129-131 at page 5 should be shifted above, at page 4 line 123, and modified as follows: "Notably, in figure 2 prominent connections between plasma and serum apelin concentrations and MADRS items 5 (reduced appetite) and 6 (reduced sleep) are reported, suggesting potential associations between apelin levels and these specific symptoms". Then:" Figure 3 shows significant Pearson correlation results as bubble graphs.
Discussion
Page 6, lines 160-161: The authors should correct the text accordingly to: "Similarly, in another clinical study, serum apelin levels were found significantly higher in patients diagnosed with depression".
Page 6, line 165: The authors should change as that: "..between blunted apelin release and MDD".
Page 6, 166-168: This sentence should be clearer, for instance as: ""This finding is intriguing, since serum apelin levels, despite being lower on average in patients, did not exhibit significant differences between the two groups under investigation, suggesting that plasma measurements might be more reflective of biological changes underlying pathophysiology of MDD". Also authors should briefly provide an hypothetical explanation of the discrepancies between serum and plasma observed in their study.
Page 6, lines 175-177 : the authors should write: "These contrasting findings highlight the complexity of apelin's physiological role as well they suggest that variation of this peptide in the bloodstream would reflect different depression conditions. Further investigation on more detailed patients' clinical presentation are needed to better understand the relationship between circulating apelin fluctuations and MDD symptomatology".
Page 6-7, lines 168-194: This part is still unclear. Without knowing the precise clinical procedure adopted is not possible to explain and interpret correlation results, without generating confusion and misunderstanding.
Page 7, lines 201-212: As a consequence of the previous comment, these sentences should be revised accordingly.
Further effort is thus required from the authors to attain a more definite and structured version of this manuscript. This will enable a better understanding of the possible use of apelin in clinical psychiatry. In my opinion, the results are not yet complete and their interpretation still needs to be carefully considered in the light of the modifications that are now required.
Author Response
Response to Reviewer #1:
The latest version of the manuscript of Chibaatar et al, entitled:"Evaluating apelin as a potential biomarker in Major Depressive Disorder: Correlation with clinical symptomatology" has been revised as suggested, and is now much more comprehensible. By this new and clearer version of the paper, it has been possible to check the presence of other issues. Indeed, some main methodological issues now emerge, still requiring explanation in order to ensure a correct interpretation of results. Also, additional analyses are recommended, if these have not been carried out, since the manuscript is not informative enough in this respect.
We sincerely express our sincere gratitude for your careful and thoughtful review of our manuscript. Your insightful and constructive feedback has been essential in improving the clarity and quality of our work. We have carefully considered each of your comments and suggestions, and we believe that the revisions have strengthened the manuscript. Below, we provide point-by-point responses, detailing the changes made in response to your recommendations.
Main points:
Material and methods
This section still requires careful revision and extension.
1) First, the adopted MADRS functioning must be clearly explained in order to be easily understood also by readers who are not directly involved in clinical psychiatry assessments, enabling a full comprehension of the statistical significance reported.
Specifically:
Page 8, lines 239-242: The authors should provide at this point the extensive explanation of the functioning of the administered MADRS scale. This is necessary to interpret correlation results.
We appreciate your valuable suggestion to clarify our methodology. To ensure clarity and accessibility for all readers, including those not directly involved in clinical psychiatry, we have provided a more extensive explanation of the MADRS scale in the revised manuscript. The following text has been added on page 8, lines 273–291. “Symptoms were assessed using MADRS, a clinician-administered instrument widely used to assess the severity of depressive symptoms in patients with MDD. It comprises 10 items, each evaluating a distinct symptom domain associated with depression. Each item is scored on a 7-point scale, ranging from 0 (absence of symptoms) to 6 (severe symptoms), with a total score ranging from 0 to 60. Higher total scores reflect greater severity of depressive symptoms 50,51. The 10 items are described as follows: 1) Apparent Sadness: Observable despondency, gloom, and despair, assessed through speech, facial expression, and posture. 2) Reported Sadness: Subjective reports of low mood and hopelessness, rated by intensity and duration. 3) Inner Tension: Feelings of discomfort, edginess, or mental tension, escalating to panic or dread. 4) Reduced Sleep: Decreased sleep duration or quality compared to the individual’s usual pattern. 5) Reduced Appetite: Loss of desire for food, requiring effort to maintain regular eating. 6) Concentration Difficulties: Impairment in focusing or maintaining attention. 7) Lassitude: Difficulty initiating and performing routine activities due to low energy. 8) Inability to Feel: Reduced emotional responsiveness and diminished interest in normally pleasurable activities. 9) Pessimistic Thoughts: Negative cognitions, including guilt, self-reproach, and feelings of worthlessness. 10) Suicidal Thoughts: Thoughts of death, suicidal ideation, or preparation for suicide.”
Indeed, following the standard uses of MADRS questionnaire, the positive Pearson correlations obtained in this work rather reflect those patients showing highest apelin plasma or serum levels also reported the highest scores (higher severity) at the MADRS total questionnaire, as well as at the MADRS 5 and 6 items.
We greatly appreciate the reviewer’s comment and acknowledge the complexity of interpreting the positive correlation between apelin concentrations and depressive symptom severity in our data. Although we observed a positive correlation between apelin levels and the severity of depressive symptoms (as measured by the MADRS), the MDD group displayed lower apelin concentrations compared to HCs. This observation might seem counterintuitive given the positive correlation. We hypothesize that the positive correlation may reflect a dynamic response to depression severity rather than a direct causal relationship. Specifically, it is possible that in more severe cases of depression, apelin may be part of a compensatory or adaptive biological mechanism, with higher levels reflecting an attempt by the body to counteract certain aspects of depressive symptomatology, particularly regarding appetite (item 5) and concentration (item 6). This could explain why higher apelin levels are associated with higher MADRS scores in MDD patients, as opposed to a simple inverse relationship.
To improve clarity, we have revised the manuscript to include a more detailed discussion of this potential mechanism and the complexities surrounding the interpretation of these results as “This positive correlation between reduced apelin levels and more severe depressive symptoms may initially seem counterintuitive, as many studies suggest that lower apelin levels are associated with depression. However, it is possible that this finding reflects a more complex biological response. Specifically, the positive correlation could indicate that in cases of more severe depression, apelin might be involved in a compensatory or adaptive mechanism aimed at mitigating certain symptoms. In such cases, higher apelin levels could reflect a biological attempt to counteract the effects of severe depressive symptoms. This hypothesis aligns with the notion that apelin’s role in depression could vary depending on the stage or severity of the disorder and may not follow a simple linear relationship.” on page 7, lines 212–221. We believe this interpretation will provide a clearer understanding of the relationship between apelin levels and depressive symptoms.
Moreover: did the authors interview also non depressed control subjects by the MADRS scale and analyzed their scores in correlations? in Table 1, results seem to indicate that they did not interviewed controls. The authors should perform this evaluation if they have collected these data.
As for HCs, to ensure the inclusion of a well-characterized group, all participants underwent assessment using the Structured Clinical Interview for DSM-5 Disorders (SCID-5) to confirm the absence of psychiatric diagnoses. The MADRS was not administered to the HCs group, as it is specifically designed to evaluate depressive symptom severity in individuals diagnosed with depression. Given the primary focus of our study – to examine the relationship between depressive symptoms and apelin levels in patients with MDD – the use of the SCID-5 was deemed sufficient to establish the non-depressed status of HCs. We acknowledge, however, the potential value of future studies exploring subclinical depressive symptoms in non-depressed populations and investigating their association with biomarkers such as apelin. This remains an important avenue for future research.
2) Drug treatment - Page 8, line 248: The authors should explain at this point that, in order to standardize patients in respect to the different treatments applied, they used Japanese guidelines for antidepressant drug equivalence, providing the dedicated link. This is important to understand Table 1 and the patients' clinical picture.
Thank you very much for highlighting the importance of clarifying our approach to standardizing treatments. We have now included an explanation in the manuscript as “There were no restrictions regarding psychotropic medication use or the frequency of hospital visits for outpatient participants. However, to account for variations in psychotropic treatment and ensure standardization in data analysis, we utilized the Japanese guidelines for antidepressant drug equivalence. These guidelines convert various antidepressant doses into equivalent Imipramine levels, allowing for a more accurate comparison of treatments across patients. This approach minimized potential variability due to differing medication regimens and facilitated a clearer understanding of the clinical characteristics presented in Table 1. The guidelines are available at [http://www.jsprs.org/en/equivalence.tables/].” on page 8, lines 297–303.
Others points and detailed comments:
Introduction
Page 2, Line 47: The authors should change "Furthermore,.." into: " In particular, .."
Thank you for your detailed suggestion and the term "Furthermore" on Page 2, Line 47 has been revised to "In particular" as recommended.
Page 2, lines 52-55: " Significant reductions in neurological deficits and infarct volume, and blood-brain barrier (BBB) protection in rat and mouse models of cerebral ischemia were observed 27,28. In addition, it reduced brain damage in mice with traumatic brain injury 29." The sentence is unclear, it should be changed as, for instance: "Apelin has been reported to significantly prevent from and/or reduce neurological deficits, infarct volume, and blood-brain barrier (BBB) damage in rodent models of cerebral ischemia27,28. In addition, this peptide has been found to restrain neuronal damage in mice with traumatic brain injury29."
Thank you for your suggestion to improve readability and precision of our manuscript. We have revised the section as “Previous studies have shown that apelin significantly mitigates neurological deficits, reduces infarct volume, and protects the blood-brain barrier (BBB) in rodent models of cerebral ischemia 27, 28. Furthermore, this peptide has been demonstrated to decrease neuronal damage in mice subjected to traumatic brain injury 29.” on page 2, lines 53-56.
Page 2, lines 57-58: The authors should rather write: "The potential effects of apelin on mood regulation and emotional behavior have been investigated, but studies conducted so far are limited".
Thank you for your more precise phrasing. The sentence has been revised as you suggested “The potential effects of apelin on mood regulation and emotional behavior have been investigated, but studies conducted so far are limited.” on page 2, lines 59 – 60.
Page 2, lines 68-69: "First, we sought to investigate the differential levels of apelin in serum and plasma of healthy controls (HCs) and patients with MDD." Comment: the authors should clearly and briefly state after this sentence why they investigated these two biological samples, explaining potential differences in apelin concentration in serum and plasma, if experimental and/or physiological, to better introduce the work. Indeed, usually serum and plasma molecular components are at similar concentrations, except for specific compounds participating in clotting processes and platelet release.
Page 2, lines 69-71: The authors should rather modify as indicated here: "Given the poor understanding of the roles of specific isoforms by now, assessing total apelin in both serum and plasma may offer a preliminary useful measure of the systemic concentration in patients with MDD..."
We appreciate your insightful comment and the opportunity to clarify our rationale. The manuscript has been revised to include a more detailed explanation as “This approach was informed by the variability in prior research, where apelin levels have predominantly been studied in either serum or plasma, with limited investigations addressing both matrices simultaneously. Additionally, studies that have analyzed both serum and plasma typically focused on specific apelin isoforms, such as apelin-13 and apelin-36, rather than total apelin levels. Given the limited understanding of the roles of specific isoforms, assessing total apelin in both serum and plasma may provide a valuable measure of systemic apelin concentration in patients with MDD. This dual analysis allows for cross-validation of findings and facilitates a more comprehensive evaluation by capturing the combined effect of all bioactive isoforms on apelin receptors. By examining both matrices, this study aims to provide a robust understanding of apelin’s biological distribution and its potential as a biomarker in mood disorders.” on page 2, lines 71 – 82.
Page 2, lines 77-79: At this point, the authors should change the manuscript as follows: "..allow to explore the potential predictive value of total apelin levels in respect to the presence of depressive symptoms as well as to their degree of severity, thereby providing insights that could guide towards considering this peptide as a biomarker in clinical settings."
Thank you very much for your suggestion to improve clarity and precision of this section. We have revised the manuscript as you recommended “This aspect of the study might reveal preliminary associations in MDD and allow to explore the potential predictive value of total apelin levels in respect to the presence of depressive symptoms as well as to their degree of severity, thereby providing insights that could guide towards considering apelin as a biomarker in clinical settings.” on page 2, lines 85 – 88.
Results
Page 3, line 91: The authors should indicate in the Table caption that Imipramine dosage reflects the equivalent dosage of the administered antidepressant drugs in patients, accordingly to national guidelines.
Thank you for pointing it out. We have added “Imipramine dosage reflects the equivalent dosage of the administered antidepressant drugs in patients, according to national guidelines.” In the Table 1 caption.
Page 3, lines 97-98: "..with a p-value of 0.002, indicating statistical significance. (Table 2 and Figure 1)". The authors should modify the text as such:"..with a p-value of 0.002, indicating statistical significance. (Table 2). Figure 1 depicts instead the same data reported in Table 2, as a scatterplot graphical representation of single values". The authors should also add a short sentence indicating that their patients resulted moderately depressed from Table 1 results.
Thank you for the insightful suggestion. We have revised the text to incorporate the recommendation as “..with a p-value of 0.002, indicating statistical significance (Table 2). A graphical representation of the same data is provided in Figure 1, illustrated as a scatterplot of individual values. Moreover, as reported in Table 1, the clinical profile of the patients in this study indicates a moderate severity of depressive symptoms.” on page 3, lines 108–112.
Page 3, lines 105-106: The authors should modify the text such as: "MDD, patients with major depressive disorder".
Thanks for your precise suggestion. We have modified all abbreviations as “MDD, patients with major depressive disorder” to improve the clarity. All modifications are highlighted in blue.
Page 3, line 109: The authors should report this sentence as: "Detailed Pearson correlation data are provided in Table 3 and Figure 2".
We have changed the sentence as you suggested on page 3, line 123.
Page 4, Table 3: The Table presentation is not clear as that. The authors should name columns as: MADRS, Apelin serum, r coefficient, Apelin plasma, r coefficient.
Thank you for highlighting it, we have revised the column names in Table 3 to clearly indicate that the values represent r coefficients, improving clarity and interpretability.
Page 4, line 123: The authors should provide a brief presentation of results presented in Figure 2 and figure 3. For instance, lines 129-131 at page 5 should be shifted above, at page 4 line 123, and modified as follows: "Notably, in figure 2 prominent connections between plasma and serum apelin concentrations and MADRS items 5 (reduced appetite) and 6 (reduced sleep) are reported, suggesting potential associations between apelin levels and these specific symptoms". Then:" Figure 3 shows significant Pearson correlation results as bubble graphs.
We agree that providing a brief presentation of the results depicted in Figures 2 and 3 at the specified location would enhance the clarity and flow of the manuscript. Therefore, we have added “In Figure 2, prominent associations between plasma and serum apelin concentrations and MADRS items 5 (reduced appetite) and 6 (reduced sleep) are reported, suggesting potential associations between apelin levels and these specific symptoms. Additionally, Figure 3 presents significant Pearson correlation results as bubble plot graphs.” on page 5, lines 137–140.
Discussion
Page 6, lines 160-161: The authors should correct the text accordingly to: "Similarly, in another clinical study, serum apelin levels were found significantly higher in patients diagnosed with depression".
The text has been revised as “Similarly, in another clinical study, serum apelin levels were found significantly higher in patients diagnosed with depression compared to healthy controls.” on page 6, lines 176–178.
Page 6, line 165: The authors should change as that: "..between blunted apelin release and MDD".
Suggested change has been made on page 7, lines 181 and now the sentence is read as “In contrast to these studies, our results showed a significant reduction of plasma apelin levels in patients with MDD, suggesting a potential link between blunted apelin release and MDD.”.
Page 6, 166-168: This sentence should be clearer, for instance as: "This finding is intriguing, since serum apelin levels, despite being lower on average in patients, did not exhibit significant differences between the two groups under investigation, suggesting that plasma measurements might be more reflective of biological changes underlying pathophysiology of MDD". Also, authors should briefly provide a hypothetical explanation of the discrepancies between serum and plasma observed in their study.
We appreciate all your suggestions to contributing to improve our manuscript. As your suggestion, we have revised the sentence as “This finding is intriguing, since serum apelin levels, despite being lower on average in MDD patients, did not exhibit significant differences between the two groups under investigation. This suggests that plasma measurements might be more reflective of biological changes underlying pathophysiology of MDD.” on page 7, lines 182–185. Moreover, we have added short hypothetical explanation as “The discrepancies between serum and plasma apelin levels may arise from differences in sample collection. Serum is obtained after clotting, which can trigger the release of bioactive molecules or influence the removal of apelin by platelets or clotting factors. In contrast, plasma measurements, taken before clotting, may provide a more accurate reflection of circulating apelin, potentially offering better insight into its role in MDD pathophysiology. However, further research would be needed to confirm the precise mechanisms behind these differences.” on page 7, lines 185–191.
Page 6, lines 175-177: the authors should write: "These contrasting findings highlight the complexity of apelin's physiological role as well they suggest that variation of this peptide in the bloodstream would reflect different depression conditions. Further investigation on more detailed patients' clinical presentation is needed to better understand the relationship between circulating apelin fluctuations and MDD symptomatology".
Thank you for suggesting more clear phrasing. We have revised the sentence as “These contrasting findings highlight the complexity of apelin's physiological role and suggest that variations in this peptide in the bloodstream may reflect different depression conditions. Further investigation into more detailed clinical presentations of patients is needed to better understand the relationship between circulating apelin fluctuations and MDD symptomatology.” on page 7, lines 198–202.
Page 7, lines 201-212: As a consequence of the previous comment, these sentences should be revised accordingly.
To address the concerns raised, we have revised the section as “This positive correlation between reduced apelin levels and more severe depressive symptoms may initially seem counterintuitive, as many studies suggest that lower apelin levels are associated with depression. However, it is possible that this finding reflects a more complex biological response. Specifically, the positive correlation could indicate that in cases of more severe depression, apelin might be involved in a compensatory or adaptive mechanism aimed at mitigating certain symptoms. In such cases, higher apelin levels could reflect a biological attempt to counteract the effects of severe depressive symptoms. This hypothesis aligns with the notion that apelin’s role in depression could vary depending on the stage or severity of the disorder and may not follow a simple linear relationship.” on page 7, lines 212–221.
Further effort is thus required from the authors to attain a more definite and structured version of this manuscript. This will enable a better understanding of the possible use of apelin in clinical psychiatry. In my opinion, the results are not yet complete, and their interpretation still needs to be carefully considered in the light of the modifications that are now required.
We deeply appreciate your thoughtful and constructive feedback. We recognize the importance of refining our manuscript to enhance its quality and clarity. Guided by your thoughtful and constructive feedback, we have implemented substantial revisions to address the concerns raised and improve the overall presentation of our findings. While this study represents preliminary research, we hope the revisions made effectively strengthen the manuscript and provide valuable insights. We remain committed to conducting more comprehensive studies in the future to build on these findings and contribute to the field.
Round 4
Reviewer 1 Report
Comments and Suggestions for Authors
This new version of the manuscript of Chibaatar et al, entitled:"Evaluating apelin as a potential biomarker in Major Depressive Disorder: Correlation with clinical symptomatology" has been now revised accordingly to all suggestions, but I recommend to modify some few points in the Abstract, Discussion and Method sections.
Abstract: line 16- The finding of positive correlations between apelin levels and depression severity scores should be better highlighted , also evidencing that this relationship was found within patients, as for instance: "Intriguingly, a positive moderate correlation was observed between patients' total MADRS scores and apelin levels in plasma (r = 0.439), with statistical significance (p < 0.05).."
Discussion: Lines 202-203: This sentence should be modified accordingly to or similarly: "Correlation analysis carried out in patients revealed a significant positive association between reduced plasma apelin levels and more severe depressive symptoms".
Lines 216-217: Since other molecules are supposed to act in a similar compensatory manner, I suggest to modify this sentence as that, or similarly: "In such cases, higher apelin levels could reflect a biological attempt to counteract the effects of severe depressive symptoms. Apelin is supposed to be part of a compensatory molecular response in subgroups of more severely depressed patients or presenting specific symptoms, suggesting to pursue the investigation also involving other peptides and trophic factors".
Line 343: please modify accordingly: "..for apelin as one of the molecular targets to consider in mood disorders".
Results: Please add the n value, or the number of subjects evaluated, in Table 3 and Figures 2 and 3.
Methods: lines 335: please change accordingly to: "Continuous variables were compared in the patients and control groups using the student’s t-test, while the relationship between apelin levels and MADRS scores was assessed in patients using partial correlation analysis, with sex, age, BMI, age at depression onset, and imipramine dosage included as covariates."
Also: Why the authors showed then Pearson's correlations ? Which statistical analysis did they follow? Please check this point and modify the manuscript accordingly to the statistical test used.
Author Response
Response to Reviewer #1:
This new version of the manuscript of Chibaatar et al, entitled:"Evaluating apelin as a potential biomarker in Major Depressive Disorder: Correlation with clinical symptomatology" has been now revised accordingly to all suggestions, but I recommend to modify some few points in the Abstract, Discussion and Method sections
Once again, we truly appreciate your thoughtful feedback and the opportunity to improve the quality of our manuscript. Your insightful comments have been invaluable in refining our study. Below, we have provided point-by-point responses to address your suggestions.
Abstract: line 16- The finding of positive correlations between apelin levels and depression severity scores should be better highlighted , also evidencing that this relationship was found within patients, as for instance: "Intriguingly, a positive moderate correlation was observed between patients' total MADRS scores and apelin levels in plasma (r = 0.439), with statistical significance (p < 0.05).."
Thank you for the suggestion and we agree that the positive correlations between apelin levels and depression severity should be more clearly highlighted, especially within the patient group. Therefore, we have revised the section as “Within patients with MDD, a positive moderate correlation was observed between total MADRS scores and plasma apelin levels (r = 0.439), with statistical significance (p < 0.05). Additionally, significant positive correlations (p < 0.05) were found between both serum and plasma apelin levels and MADRS subscales 5 (reduced appetite) and 6 (concentration difficulties).” on page 1, lines 17–20.
Discussion: Lines 202-203: This sentence should be modified accordingly to or similarly: "Correlation analysis carried out in patients revealed a significant positive association between reduced plasma apelin levels and more severe depressive symptoms".
Thank you for your suggestion to clarify the section. We have revised the sentence as
“Correlation analysis carried out in patients with MDD revealed a significant positive association between reduced plasma apelin levels and more severe depressive symptoms, as measured by the total MADRS score.” on page 7, lines 203 – 205.
Lines 216-217: Since other molecules are supposed to act in a similar compensatory manner, I suggest to modify this sentence as that, or similarly: "In such cases, higher apelin levels could reflect a biological attempt to counteract the effects of severe depressive symptoms. Apelin is supposed to be part of a compensatory molecular response in subgroups of more severely depressed patients or presenting specific symptoms, suggesting to pursue the investigation also involving other peptides and trophic factors".
Thank you for your valuable suggestion. We have revised the sentence to incorporate your point and emphasize the role of apelin in a broader compensatory response alongside other molecules. The revision is made as “In such cases, higher apelin levels could reflect a biological attempt to counteract the effects of severe depressive symptoms. Apelin is hypothesized to play a role in a compensatory molecular response, particularly in subgroups of patients with more severe forms of depression or those presenting specific symptoms. This suggests that further investigation is needed, not only to explore apelin's role but also to include other peptides and trophic factors that might contribute to this complex response.” on page 7, lines 218 – 223.
Line 343: please modify accordingly: "..for apelin as one of the molecular targets to consider in mood disorders".
We have made changes as you suggested, and the revised sentence now reads “The observed elevation in plasma apelin levels and their significant correlation with the severity of depressive symptoms provide compelling evidence for apelin as one of the molecular targets to consider in mood disorders.”
Results: Please add the n value, or the number of subjects evaluated, in Table 3 and Figures 2 and 3.
Thank you very much for highlighting this. We have added number of the subject in the first row of the Table 3 and the legends of the Figure 2 and 3.
Methods: lines 335: please change accordingly to: "Continuous variables were compared in the patients and control groups using the student’s t-test, while the relationship between apelin levels and MADRS scores was assessed in patients using partial correlation analysis, with sex, age, BMI, age at depression onset, and imipramine dosage included as covariates."
Thank you very much for suggesting a more clarified version. We have revised the sentence as “Continuous variables were compared in the patients with MDD and HCs groups using the student’s t-test, while the relationship between apelin levels and MADRS scores was assessed in patients with MDD using partial correlation analysis, with sex, age, BMI, age at depression onset, and imipramine dosage included as covariates.” on page 9, line 333–337.
Also: Why the authors showed then Pearson's correlations ? Which statistical analysis did they follow? Please check this point and modify the manuscript accordingly to the statistical test used.
We sincerely apologize for the confusion caused by the error in our manuscript. All correlation analyses were conducted as partial correlations with sex, age, BMI, age at depression onset, and imipramine dosage included as covariates. We have revised the manuscript to accurately reflect the statistical test we used.
Once again, we would like to express our sincere gratitude for the time and effort you have dedicated to reviewing our manuscript. Your thoughtful comments and constructive feedback have been invaluable in improving the quality of our work. We truly appreciate the opportunity to enhance our manuscript and are confident that the revisions have strengthened the overall clarity and rigor of the study.